# He Karanga Maha. Investigating Relational Resource Management in Aotearoa, New Zealand

**DOI:** 10.3390/ijerph20085556

**Published:** 2023-04-18

**Authors:** Sarah Rewi, Daniel Hikuroa

**Affiliations:** 1School of Biological Sciences, Waipapa Taumata Rau, The University of Auckland, Auckland 1023, New Zealand; 2Department of Māori Studies, Waipapa Taumata Rau, The University of Auckland, Auckland 1023, New Zealand

**Keywords:** indigenous harvest, natural resource management, seabirds

## Abstract

Reciprocity amongst Māori peoples and the natural world is the foundation of the Māori worldview and natural resource management. Autonomy over resource management and the associated practices is an essential component of Māori wellbeing. This paper investigates the cultural, spiritual, historical, and ecological dimensions of mutton-bird harvesting, to gain a better understanding of the relational approach of Māori natural resource management. Resource management in Aotearoa New Zealand currently lacks the relational approach seen in Māori customary harvests. Therefore, the objective of this study is to identify the key values that underpin this cultural practice. Semi-structured interviews identified three key themes: harvesting practices, kaitiakitanga (resource management based on a Māori worldview), and whanaungatanga (kinship between people). Harvest practices had a bottom-up governance approach creating diverse harvesting techniques that adapt to local environments. Kaitiakitanga identified mana whenua rights to decision-making power in natural resource management as a requirement for success. Whanaungatanga also identified relationships and collaboration as a vital component. To optimize the best outcomes for the environment, we advocate for a genuine cross-cultural and relational approach and the inclusion of these practices and values in the governance of natural resources in Aotearoa New Zealand.

## 1. Introduction

For over a thousand years, Māori have lived with the environment guided by the responsibilities promoted by our ancestors to maintain the balance amongst humans and the natural world [1]. Thus, the lands and waters under Indigenous care thrive as an economic, cultural, and spiritual foundation for all Māori people [2]. Although many definitions for Indigeneity exist, a commonality of Indigenous worldviews is the emphasis of the relationship between humans and the natural world. Understanding this relationship is valuable because cultural revitalization and Indigenous governance policies improve Indigenous well-being. Although Indigenous affinity with the environment is often perceived as a political stance due to its association with Indigenous governance, strong correlations have been found between ‘nature relatedness’ and Indigenous culture [3].

Given this, it is unavoidable that the degradation of this relationship and exclusion of Indigenous environmental management negatively impacts Indigenous health and well-being [4,5]. For Indigenous peoples, these relationships do not merely shape reality, they are reality [6]. For Māori, the nature of this reality is whakapapa, the genealogical ties that connect people with each other and the environment [7]. These relationships between humans and all things, invoke reciprocity, responsibility, and respect. Although often regarded as a philosophical approach, this relational ontology is essentially practical by regulating human interactions [8,9]. 

In Aotearoa New Zealand, colonial policies and governing systems have marginalized Māori ways of knowing in favor of Western science for the past 100 years. Consequently, most research has not incorporated Māori practices, people, values, or knowledge [10]. However, in more recent times, Māori engagement with research and research institutions is gaining momentum. Western scientific disciplines such as environmental studies and ecology, are showing an increasing interest in Māori knowledge and ways of knowing due to the differing perspectives they provide [10,11]. Moreover, the knowledge systems held by Māori have uniquely evolved to New Zealand’s endemic environmental context and therefore provide historical and context-specific insights to natural resource management [12].

Māori resource management practices are often described as kaitiakitanga, deriving its meaning from the word ‘tiaki’ which, depending on context, can translate as ‘to guard’, ‘to preserve’, ‘to keep’, ‘to protect’, ‘to foster’, ‘to shelter’, and ‘to conserve’. The prefix ‘kai’ refers to the agent of the action meaning a kaitiaki is a protector, guardian, preserver, keeper, conservator, and foster-parent [13]. Traditionally, kaitiaki are spiritual agents of the gods, including ancestors, that protect the natural elements including the seas, flora and fauna, sky, rain, winds and storms, volcanic activity, and people. These resources are taonga (treasures) that are greatly cherished. Kaitiaki manifest in forms such as fish, animals, trees, or reptiles to guide human action to maintain balance with the environment [14]. The suffix ‘tanga’ subtly changes the meaning to reference the broader aspects of an action. Hence kaitiakitanga means trusteeship, preservation, and guardianship [13,15].

Despite being a modern term, the principles and values which underlie kaitiakitanga are centuries old. Seeking balance with natural resources is not only necessary for Māori economic survival but is an inherent obligation to the wellbeing of future generations. It creates a whakapapa link between the past and the future, the spiritual and the human realms, and to ensure long-term survival, the environment must be protected [1]. The protocols and rituals that guide these efforts are unique to each iwi (Māori tribal nation), hapū (a kinship group within an iwi), whānau (family line within a hapū) and hapori (community). These are often interpreted as sustainable management but without reference to whakapapa, kaitiakitanga cannot be understood [15]. Whakapapa connects Māori to kaitiaki and through them ensures that the mauri (life force) of the environment remains healthy. As such, every hapū are kaitiaki in their tribal regions in which they hold mana whenua status through this whakapapa link [14]. Consequently, kaitiakitanga is relational in its approach, making it the obligation and responsibility that lies with mana whenua: those who maintain an Indigenous relationship to the land. This understanding is echoed is the words of Apirana Mahuika of Ngāti Porou:

“Nōku tēnei whenua, kei a au te kōrero. Nōku tēnei whenua, ko au te Rangatira. This is my land; this is my story to tell. This is my land, and I am the authority”.[2]

Kaitiakitanga is intertwined with other key concepts including mana (authority), tapu (sacredness), manaaki (hospitality), and tuku (gift, transfer), encompassing a vast world of Māori traditions and customs [13,15]. It is therefore inappropriate to merely define kaitiakitanga as ‘guardianship’, as the Crown and the New Zealand Government have. ‘Stewardship’ is another common definition that does not accurately reflect the philosophies of kaitiakitanga [15]. Both interpretations diminish the complexity of kaitiakitanga and place western ideals of human superiority and ownership over the human–environment relationship, which contrasts with a Māori worldview [13].

Customary harvest practices play a crucial role in Indigenous resource management and have significantly contributed to global biodiversity [16]. In Aotearoa New Zealand, mutton-bird harvests have been conducted for nearly a millennium as a tool of survival and resource management [17]. If kaitiakitanga successfully stimulates the mauri of a resource, and/or ensures it flourishes, it provides a bountiful harvest that becomes freely available to the people [18]. Customary harvests demonstrate an intimate knowledge of migration, breeding cycles, and feeding habits: all information gathered from the environment through generations of intimacy with the natural world [18]. These relationships regulate human interaction with harvests to minimize wastage and overharvesting, which is crucial for the appropriate delivery of harvesting practices [19].

It is well acknowledged that engaging in traditional practices holds a cultural significance that contributes to Indigenous people’s well-being [20]. Throughout history, Māori mutton-bird harvests have maintained a connection to both people and place. They have been the foundation for many inter-tribal relationships and, from a Māori understanding of the world, the interconnection between humans and nature [21,22]. The ability of many iwi to legally conduct harvests or access to their traditional birding grounds has been prohibited through modern-day colonial resource practices and policy. Therefore, undertaking these practices is an important expression of cultural revitalization for the benefit of Māori wellbeing and relationships.

Asserting autonomy over environmental management upholds these practices within the community. This paper investigated the cultural, spiritual, historical, and ecological dimensions of mutton-bird harvesting to gain a holistic perspective of the Indigenous management of these birds. Our objective is to identify key themes and values associated with customary harvests as a relational resource management approach.

## 2. Materials and Methods

This study was conducted incorporating a kaupapa Māori approach, exercising cultural practices, ethics, and customs. In Aotearoa New Zealand, semi-structured interviews have been favored by previous studies for collecting information around customary harvests of two mutton-bird species—ōi (grey faced petrel, *Pterodroma gouldi*) and tītī (sooty shearwater, *Puffinus griseus*) (see [17,20,21,22]). A glossary of these terms can be found in the Appendix A. Given the success of this method in effectively engaging with Māori communities, worldviews, and knowledge, semi-structured interviews were the chosen method for qualitative data collection. Questionnaire design included open-ended questions pertaining to mutton-bird harvests from the literature.

Interviews were conducted with two participants. The first is a kaumātua (elder) from Ngāti Wai who is an identified knowledge holder in Northland, New Zealand, and participated in cultural harvests of mutton bird as a child. The second participant was a Trustee representative from Te Kawerau ā Maki who manages the environmental and cultural heritage portfolio for the iwi and whose parents, uncles, and aunts practiced mutton-bird harvests. These participants are also recognized knowledge holders. The knowledge held by each represents the combined experiences and observations made through multi-generations, and thus reflects the voices of many. For the last 2–3 generations, muttonbirding in the regions of the participants has been illegal and/or the local communities have been prevented from accessing the muttonbirding sites (i.e., the selling of land to private landowners). As such, there is an urgency to capture the few remaining community members with direct experiential knowledge of these traditional practices. Therefore, despite the number of participants being small, the quality of their perspectives is immeasurable and offers a chance to understand the details of their region-specific muttonbirding practices.

All semi-structured interviews were conducted under the University of Auckland Ethics Approval permit reference number UAHPEC22193. Selection of participants was guided by pre-existing relationships, prioritizing those who have an intimate connection with cultural harvests. Three interviews per participant were conducted, including informal discussions based around relationship building and rapport. Interviews were approximately 45 minutes in length and were led by the participants. Due to the COVID-19 pandemic, two interviews were conducted via online video conferencing; however, these took place after the in-person interviews. Discussions were audio recorded and transcribed for analysis. Transcripts were reviewed and approved by each interviewee before the final analysis was conducted.

Thematic analysis was used to analyze the data for common patterns or themes between participants. This form of analysis is a ‘bottom-up’ research tool that provides a detailed account of data by separating narrative materials into smaller units that can be analyzed against other narratives and the literature [23]. As Māori customary harvests varied significantly between regions, thematic analysis was the method chosen to interpret the interviews. We aimed to determine the common themes and values associated with the application of past mutton-bird harvests. A mixed-method design that combined a code book approach (as described by Gale et al. [24]) and a reflexive approach (Braun and Clarke [25]) was used for analysis. This approach is deductive in the interpretation of participant transcripts wherein themes based on existing theoretical research have been identified. Transcripts were coded according to underlying themes, emotions, and elements (e.g., recounting of events, personal experience, particular behaviors) to classify passages by importance. A set of potential themes from the pre-existing literature were then compared with the transcript codes for overlapping similarities (see [1,13,14,15,16,19,26,27,28]).

## 3. Results and Discussion

Thematic analysis of the interviews produced three themes: harvest practices, kaitiakitanga, and whanaungatanga, all centering whakapapa as a relational foundation. By analyzing these themes, we can begin to understand the customs, principles and ethics through which Māori engage with mutton-bird harvests as a relational resource management practice.

There is a need to investigate Māori approaches to resource management to sustainably adapt current natural resource management practices to modern-day environmental challenges. For instance, Aotearoa New Zealand is a global hotspot of seabird diversity, hosting approximately one quarter of the world’s seabird species with 10% being endemic to New Zealand breeding grounds [29]. The value of these species as ecosystem engineers is recognized as a key restoration tool for native ecosystems [30]. Yet, these seabird populations are in decline. Twenty-three of New Zealand’s thirty-six endemic seabirds are classified as ‘vulnerable’, ‘critical’, or ‘endangered’ according to the IUCN (International Union for Conservation of Nature) criteria.

Previous studies on mutton birds solely focus upon quantitative investigation wherein bird diet, behavior, morphology, physiology, and ecology are explored (see [27,29,30,31,32,33]). Notably the work of Lyver et al. [17] and Lyver and Moller [19] have incorporated Māori perspectives using a more qualitative approach. They have explored the role of Indigenous knowledge within contemporary natural resource management and the use of customary harvesting practices for safeguarding seabird populations [34,35]. This paper looks to expand on their work by providing perspectives from differing tribal nations as to prevent the homogenization of Māori harvesting practices.

### 3.1. Harvest Practices

For many Indigenous peoples, resource harvesting practices are built upon the extensive behavioral and ecological knowledge of their own localities, accumulated over generations. Consequently, such practices enhance biodiversity and reflect the strong connection between Indigenous peoples and the natural world [26,36]. For Māori, traditional guardianship of mutton-bird species also includes customary harvest, which is embedded within a cultural and spiritual context [19]. Customary management of ōi has sustained colonies through harvesting practices and restrictions. These include the harvesting of pre-fledged chicks, the prohibition of damaging burrows, and the rotation of harvests between populations. Consequently, customary harvest practices have little impact on population growth rates even when up to 75% of chicks are harvested [16]. Harvest rates of tītī have also proven to be sensitive to a colony’s chick density. It therefore can provide an efficient method of monitoring population trends which is an important component of sustainable long-term harvests [19]. This demonstrates how the implementation of cultural practices is beneficial to sustainable resource management.

Harvesting practices were a significant theme throughout the semi-structured interviews, with each participant describing them in detail. Participant 1 described ōi harvests from their childhood undertaken by Ngāti Wai on the Whangārei coast:

‘In my youth I remember the mutton-bird season running from September to October. From my very earliest memories, harvesting would happen on the mainland as well as the islands out on the Taiharuru river near Pātaua and the Whangārei heads. By the time I was active in the harvesting, the birds had shifted off the mainland. So, most of my experiences with the mutton-bird harvest were on the three islands near Taiharuru. Prior to collecting the ōi, each morning would begin with karakia which included a dedication to the ōi harvest. My brothers would get lowered down the side of the cliffs tied to a rope and to where the mutton-bird holes were. The chicks looked like balls of fluff. To catch the birds, they would reach inside the burrow with a forked stick, try to trap the head and then reach in, their hand protected by a sugar sack, and break the neck. If they were pulled out into the light, they would more often vomit, which would spoil the meat’.(Participant 1)

Described here is the ‘forked-stick’ method of harvesting ōi chicks from their burrows. It demonstrates a comprehensive knowledge of the chick’s defensive behavior and how that impacts the state of the chick. The forked-stick method has also been recorded as a long-standing method in the Bay of Islands that is still used as one of the few legal harvesting techniques [37,38]. There is also mention of the restricted harvesting season (being September to October) which correlates with their migration and breeding patterns [39,40]. 

Another interesting note is the use of sugar sacks to protect the hands of harvesters. Use of new material shows how traditional or customary practices can be adaptive and incorporate new knowledge, technologies, and ideas. Participant 2 described ōi and tītī harvest from west Auckland in the region of Te Kawerau ā Maki undertaken by their parents’ generation:

‘My father still remembers being lowered down the cliffs at Te Henga [on a harakeke rope] to gather the mutton birds. With a kete [woven bag] on his back, he would flick what he called a bracken fern into the hole and catch the fluffy down feathers of the bird on the fern frond. Then he would pull it out and put the bird and the fern into his kete’.(Participant 2)

Here a different harvest practice is described using fern fronds rather than a forked stick to trap ōi and tītī chicks and pull them from their burrows. Another key difference was the killing of the birds themselves. Participant 1 emphasized the importance of killing the birds before they were retrieved from their burrows, whilst participant 2 described the chicks being pulled from their burrows alive. Notable similarities between each account were the targeting of pre-fledged chicks from mainland cliffside colonies reflecting how mutton-bird harvest practices can be both similar and vary between regions.

Cultural protocols are embedded within harvest practices made evident in discussions around the use of karakia. Participant 1 also emphasized the normality of karakia in not only management but everyday tasks. This is certainly an element that is missing from much modern conservation practice. Consequently, much of the older practices are being lost as Māori communities become more and more disconnected to their resources:

‘My old people would karakia the season, but later, after they had passed on, I don’t know how well those practices kept, I think they died out a bit. I don’t recall my father doing it, although he may have, and I didn’t take any notice. But I don’t recall him doing karakia before they went over to get the birds from those islands’.(Participant 1)

However, discussions with participant 2 demonstrated that this disconnection is not necessarily a barrier to revitalizing cultural practices. Aspirations for culturally significant practices around natural resources are inherited from previous generations which is more than enough to reconnect with traditional practices.

‘Mutton bird wasn’t something that I grew up eating, I was only introduced to them later in life. I have more of a wairua [spiritual] connection to the tītī based on the stories that my father tells so often. For me, it’s about being able to collect taonga traditionally from your whenua and revive those cultural practices that were happening 60–70 years ago. Whether it’s with the mutton bird or māra [garden], so long as the generations that come after us can participate in those practices of gathering kai for their whānau from their whenua’.(Participant 2)

Revitalization of customary harvesting practices is an ambition for many Māori communities. Stories passed down about these cultural practices are powerful ways in which younger generations can maintain a connection with their whenua, even when they have been physically disconnected. Te Au [41] describes a similar experience with the tītī harvests on Taukihepa, one of the Rakiura mutton-bird islands: ‘Laying eyes on Taukihepa was like a powerful magnet pulling me into its spirit. Tears welled in my eyes, from all my father’s kōrero (stories), I felt I was finally home.’

When discussing Ihumoana, a current ōi colony within Te Kawerau ā Maki’s tribal region, Participant 2 had no knowledge of ōi harvests at this site:

‘When dad talks about the stories, Ihumoana is more of a kaimoana gathering place. The cliffs at the south end of Te Henga and at the north end of Kauwahaia was where he speaks about the mutton-bird harvests’.(Participant 2)

This may be an indicator that the colonies have relocated themselves along the Te Henga over the last century, after Te Kawerau ā Maki’s occupation of Ihumoana as a fortified pā site [42]. An interesting similarity between the participant’s anecdotes of their muttonbirding stories is the involvement of young children (6–7-year-olds) in harvesting practice. Their involvement does not appear to be passive, with children having specific roles that contribute to the efficiency of harvest practices. By engaging with their whānau (extended family) and whenua (land) in this way, children develop an ecological literacy grounded in a deep sense of connection [43].

‘On the harvest, when the old people would talk about how we are connected to everything is where I got a more intimate understanding of my relationship to the bird itself. Colonies were not always abundant on those islands. I recall my father admonishing this uncle of mine for going to the islands too soon’.(Participant 1)

This fosters an intergenerational relationship between Māori and mutton-bird species by passing on values, practices, and knowledge to the younger generations. With this intimacy, communities remained attuned to the resource state which in turn guides harvesting decisions. For mutton-bird harvesting, this typically applies to decisions around the size of a harvest or if it would occur at all [19,44]. Restriction in terms of access to resources is a significant component of customary harvest practices. They ensure the long-term prosperity of a resource and maintain sustainable resource management practices. For Māori, the practice of rāhui is a common form of resource restriction that prevents the overexploitation of a specific resource or environment [13,17]. Participant 1 describes how this is incorporated into their experiences of ōi harvests:

‘Those birds that were harvested were the chicks, they wouldn’t take all of them and the old people would say which of the holes to go to and which were to be left. This was to allow the bird to mature and return to the island in the same way its parents did, for the continuation of the colony’.(Participant 1)

In modern times, the Westminster legal system is the basis of law in Aotearoa New Zealand. Thus, enforcing tikanga law, i.e., the law of rāhui, can be very difficult for many Māori communities, hapū, and iwi attempting to manage their resources under the ethic of kaitiakitanga. Participant 2 commented on the importance of this:

‘We have many aspirations, the main one obviously is to get our people back on their whenua but then there’s also that systemic connection. Legislation is great in theory but in practical terms, nothing gets done. Te Kawerau ā Maki are trying to formalize a Forum with the power to protect and regenerate resources with both rāhui and legal mechanisms’.(Participant 2)

Others have attributed European colonization as a major obstacle to the application of tikanga and the associated kaitiakitanga practices [19]. The Crown’s enforcement of laws around customary harvests, whether it be kereru, kaimoana (seafood) or mutton bird, creates an environment where tikanga and kaitiakitanga management practices are not always acknowledged.

### 3.2. Kaitiakitanga and Kaitiaki

As an ethic, value, and philosophy, kaitiakitanga guides Māori relationships and practices with the natural world. Thus, having kaitiakitanga emerge as a theme from interviews is unavoidable. Spiritual guidance of kaitiaki regulates human interactions that helps minimize wastage when conducting customary harvests [19]. As such, harvesting practices are a practical implementation of kaitiakitanga.

Kawharu [15] discusses the social dimension of kaitiakitanga and resource management. An extension of this is the social obligations of kaitiakitanga which was a very prominent theme throughout the interviews. Participant 1 conceptualized these obligations as something that is determined through whakapapa:

‘For my hapū, in these modern times, the continuation of kaitiakitanga is realized through our Trust as a parliamentary legislative body. Yet this is only important in regard to colonial law. The real kaitiakitanga rests with the hapū and our trustees see themselves as a continuation of the generations of kaitiaki going back through time. Even in terms of the treaty claims process, kaitiaki obligations going to the wider iwi is not seen as correct, as they are not mana whenua, tangata whenua, or the people of that place’.(Participant 1)

This connection between people and place has led to the creation of kaitiakitanga practices that are specific and localized and therefore vary between communities. These obligations are inherited from our tūpuna (ancestors) and are handed down to our mokopuna (grandchildren/future generations) to sustainably manage the environment throughout time [14]. Kaitiaki, as agents of kaitiakitanga, also vary in their nature between regions. This is because the connection between kaitiaki and tangata is based upon whakapapa and is therefore unique to each hapū.

‘From our perspective, we as humans are the kaitiaki. We’re the ones that have the ‘tangata’, the ability to carry out the karakia, rituals, and rites to guard the mauri of a resource to ensure that it remains in place. However, this cannot be done without the assistance of the spiritual guardians, the tupua, the deities, the atua. For us, our spiritual guardians take many forms, they have the ability, like Maui, to change shape. This is why we have a tohunga, as they have the ability to communicate at that level. Moreover, without a connection to a resource, through whakapapa and history, the work of kaitiakitanga cannot be undertaken. Therefore, it is whakapapa and our collaboration with the spiritual kaitiaki that validates our claim that we are kaitiaki’.(Participant 1)

Participant 2 describes kaitiaki through their connection to Te Henga and its spiritual, cultural, and historical importance to Te Kawerau ā Maki:

‘For us, taniwha are our kaitiaki. It is interesting how most people think of scary monsters when we say taniwha, that is so far from how we talk about them. These kaitiaki appear in the form that you need when you are ready to see them. Sometimes kaitiaki appear as a log on the lake Wainamu that’s going in a different direction to the current. There is another kaitiaki out there that takes the form of a heron, which sits on the rock in the middle of the Waitākere River. When you see those things, you know your tupuna are with you and that is a good sign. Tāngata are a kaitiaki through their own gifts, in whatever form they are. Some of our people are kaitiaki in terms of their mahi on the whenua. Others are more kaitiaki for the emotional side of people. Then you’ve got the other kaitiaki that are hard and honest that hold people to account’.(Participant 2)

It is commonly noted that if kaitiakitanga successfully stimulates the mauri of a resource, it will continue to provide a plentiful harvest [18]. Acting as kaitiaki acknowledges a region’s mauri (spiritual essence) and history. For Te Kawerau ā Maki, conducting these practices at Te Henga is central to the mana and identity of their people [45]. Diversity of thought around kaitiakitanga and kaitiaki is also expressed throughout the literature. Another Te Kawerau ā Maki member, Te Warena Taua, describes kaitiakitanga responsibilities being associated with whakapapa and mana relationships which exist irrespective of the ownership or sale of land [46].

Many misconceptions around kaitiakitanga and kaitiaki exist and these misunderstandings can lead to friction between kaitiakitanga-based and conservation-based resource governance. Although there is a role for those without this whakapapa connection to assist in kaitiakitanga, both participants acknowledged this disconnect between the differing cultural management approaches:

‘Many attitudes toward kaitiakitanga make claims to the guardianship of the environment and natural resources across the entirety of New Zealand. This perspective does not stop to accommodate the local Indigenous communities which creates a sense of intrusion that ultimately erases the people of the land in matters concerning the land. In my understanding, kaitiakitanga is a responsibility that lies solely with mana whenua and can only belong to the people of that place. All others can only assume to support them in that role. Kaitiakitanga practices over resources extend over many generations, with each successive generation stepping forward to fulfil that role’.(Participant 1)

‘I think the term kaitiaki is thrown around in the mainstream but it’s a concept that needs to be examined a lot deeper. I don’t think that you need to be Māori to be a kaitiaki; however, your form of kaitiakitanga would look different compared to Māori’.

(Participant 2)

Some describe kaitiaki as the spiritual assistants of the atua (gods), including ancestors, who oversee the elements of the natural world. Māori act as minders for their kaitiaki, who are their relations, to ensure the mauri of the natural world remains healthy and strong [14]. Other interpretations emphasize the responsibilities and weight of the leadership required to take on the role of a kaitiaki. One account by Te Au [41] articulated this especially on his kaitiaki journey: ‘I wondered whether I wanted to be a kaitiaki, with all the responsibility and strife but as one leader passes on, their mantle continues in one of their descendants to be chosen by the extended family. My old people chose me to carry the burden and under Māori law that is full and final’.

As both a concept and practice, kaitiaki and kaitiakitanga are extremely complex and dynamic which can lead to difficulties in cross-cultural communication. Attempts have been made to incorporate these terms into policy and practice such as the Resource Management Act [47]. Within this act, kaitiakitanga refers to the exercise of guardianship by the tangata whenua of an area in accordance with tikanga Māori in relation to natural and physical resources. However, absent from this legislation is the understanding of how Māori view kaitiaki and kaitiakitanga, or how it is practiced at the community level. This disregards the historical relationship between mana whenua and their environment leading to cultural harm, and the erasure of Indigenous resource management practices.

### 3.3. Whanaungatanga

From a Māori understanding of the world, humans and nature are interconnected [48]. This ideal is recorded extensively throughout the literature, particularly in discussion around customary harvests, natural resources, and Māori management practices. However, both interviews raised another dimension to this discussion that is much less obvious in the literature—the connection between humans. This plays an important part in Māori resource management practices. It is known as ‘whanaungatanga’ which describes the kinship that is held between Māori people and Māori communities. Kaitiakitanga is often described in the context of human and nonhuman relationships. However, kaitiakitanga finds its centrality in Māori kin-based communities because it weaves together ancestral threads of identity, practice, and purpose [15]. These community relationships are dictated by whakapapa and are thus held together over past, present, and future generations [8].

‘Even though kaitiakitanga is the responsibility of the mana whenua, they would never prevent the sharing and wider use of hapū resources, because their use isn’t just about food and survival. It is about an obligation to share those resources to keep alive the threads of whanaungatanga’.(Participant 1)

Sharing of resources between communities is a core practice of kaitiakitanga which influences cultural values such as manaakitanga (hospitality). A community’s status as kaitiaki for their region is defined by their ability to provide their resources to visitors. This act reflects their ability to maintain a prosperous resource and not doing so is considered shameful [19]. Participant 1 also described how the harvesting of resources was carried out collectively, where many communities would come together and share both the work and spoils of each harvest:

‘For the harvesting of ōi, families would travel to Taiharuru, Whangārei where some of the old people lived, and spend the night before going out to harvest. Kuia [elders] would keep the campfires burning all day and part of the children’s job was to make sure there was enough wood for them to cook. That was where stories would be told, stories of fishing expeditions, of the mountains and of the ancestors from Hawaiki. So those community harvesting events were not just about the food. They were about connecting to the people living in those places and connecting to our whanaungatanga, through the sharing of resources. These were times of comradery and whanaungatanga that sustained the relationship between the coastal and inland peoples’.(Participant 1)

These discussions are not uncommon, with examples of collaborative harvest and resource sharing being extensive in the literature. People from Ihumatao would travel to Ōkāhu Bay to collect pipi (*Paphies australis*), and over to Māngere to harvest pūpū (*Turbo smaragdus*). They shared those foods as it was only right and typical to share them once gathered because of their connections with both people and the whenua [46]. During the 1850s in the Hawke’s Bay, Māori would travel from Taupo and stay for about six weeks to assist in the harvesting and preservation of tītī. They brought with them five or six gourds that were packed with about 100 birds each to be taken back to Taupo [37]. Similarly, in Northland, there is an ethic of support between hapū on the east and west coasts forming inter-hapū regional alliances for major projects such as fishing, harvesting, and gardening. For instance, many hapū gathered at the Taiāmai gardens, to cooperate at the time of harvesting [38].

So, in addition to the practicality of food gathering and the transfer of skills to younger generations, gatherings for harvests also strengthen relationships between communities by supporting the long-distance flow of information [38]. Participant 2 also emphasized this collective aspect to resource management:

‘I see it being a collaborative space with an absolute shared power that uses different lenses, an inclusive process. It is still about reo and tikanga and sharing those stories of empowerment, but ultimately, we are about looking after the people. Māori and Pākehā. There are many non-Māori living at Te Henga, that we have a good relationship with. It’s only a small vocal group that aren’t willing to work with us and their responses are out of fear and ignorance’.(Participant 2)

## 4. Conclusions

A substantial gap currently exists between consultation and collaboration with Māori and sharing of decision-making power for governance of resources. Divergent philosophies, the impoverishment of Māori communities, and reluctance to share power are some of the many obstacles that have contributed to this [49]. Current resource management practice in Aotearoa New Zealand reflects a conservation-based philosophy that diverges from the social dimensions of relational kaitiakitanga-based resource management [15,28]. This paper calls for a greater appreciation of Māori cultural practices, such as mutton-bird harvests, as a relational approach to natural resource management.

Discussions of mutton-bird harvesting have identified three core themes: harvest practices, kaitiakitanga and kaitiaki, and whanaungatanga (guardianship of natural resources; territorial rights of the local Indigenous community; and relationship through kinship and shared experiences respectively [50]). The many connections among individuals, communities and the natural world reflect a relational approach to natural resource management. For customary harvests, these relationships guide the connection between humans and the mutton bird, which provides the ethical basis that underpins harvest practices and engagement with the environment. They also determine the right of mana whenua to act as kaitiaki and the obligations that are associated with this role and maintain these practices over generations to ensure the continued prosperity of the environment throughout time. Harvests not only incorporated cultural and spiritual practices but were found to be adaptive and incorporate empirically gathered ecological knowledge. It is not more data that is needed but a change of worldview that acknowledges the reciprocity between humans and the natural world [44].

Many collaborative efforts in the resource management space do not incorporate power-sharing or resource co-governance, or the acceptance of Indigenous practices. There is a necessity for collaboration and whanaungatanga to achieve culturally appropriate management practices such as the customary harvests of mutton birds. However, without legislated co-governance and/or incentivized co-management models, power sharing and equitable decision making remains an obstacle to genuine collaboration and whanaungatanga between Māori communities and mainstream conservation parties [48]. By understanding our connections to each other and the natural world, we can position ourselves in relation to our environment, both social and physical. This provides a hopeful foundation for the management of mutton birds in Aotearoa New Zealand and an incentive for the revitalization of customary practices throughout the country.

## Data Availability

Not applicable.

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
