# Peer review of "He Karanga Maha. Investigating Relational Resource Management in Aotearoa, New Zealand"

_ijerph, 2023, doi:10.3390/ijerph20085556_

Round 1
Reviewer 1 Report
First, I thank the authors for this intersting study. Studies that evaluate practices from traditional / indigenous people to incorporate their values under more contemporary practices of management and conservation are not only welcome but necessary.
My main dificulties are regarding the methods used for analysing the semi-structured interviews (thematic analysis). I know that in anthropology and sociology, qualitative/descriptive approaches are very common, but it makes difficult for a reader to link findings with the objectives. I am an ecologist, therefore I am used to quantitative approaches. I can not evaluate this study from an Anthropology/Sociology view, which is the one I believe would be better. However, I can say that the ms moves in the direction of reaching the objectives as it identifies key themes and values, but some issues put barriers to its full understanding by a reader.
The findings are obscured by the vast amount of local vocabulary, for instance, in conclusions, all the identified values are traditional terms. While there are descriptions of those terms in the text, a table summarizing the meaning of each value would make it easier for the reader to understand the final conclusions.
In the methods there are no mentions of the number of interviews, duration of interviews, description of interviewed people (i.e. frequency of sexes, range of age of the interviewed, occupation, role on the communitity, region, etc.). A supplemental material with that summary, and even with the transcripted interviews, would be very helpful.
Finally, I really would suggest the authors consider incorporating some quantitative approach to the thematic analysis, like, frequency of occurence of each value over all interviews, per interview, per region, per age, per sex, and so on.
I hope my comments are useful!
Kind regards
Author Response
Firstly, we would like to thank the reviewer for their time spent critiquing our manuscript. In particular, we appreciate the insights provided from a more quantitative-focused discipline that have highlighted a lack of clarity between our objectives, methods, and results. In the edited manuscript we have addressed this by altering the wording of our objectives to be values-focused and aimed to improve the reader’s understanding of the link between values and practice.
Supplementary material including the requested glossary table has been added. Further details have regarding the interview process has been included into the text along with details of the participants. We decided against adding a more in-depth description of each participant in the supplementary materials (e.g., gender, age, and occupation) as we felt these details are not relevant to their knowledge surrounding mutton-bird harvests. For example, harvesting practices were not age or gender specific.
Regarding the reviewer’s request to incorporate quantitative analysis into the manuscript we have discussed this and do not feel it is appropriate for this study. Our qualitative approach was chosen for the following reasons:
- Semi-structured interviews: the two-way communication provided by this method gave participants the ability to explore areas of the subject that they deemed necessary or important. It encouraged a two-way communication that was appropriate for their level of expertise which was far higher than any members of the research team.
- Thematic analysis: allowed for the incorporation of categories unknown to the researcher’s prior to the research. It also allows comparison with other literature related to Indigenous environmental practices and theories.
Other examples of this can be seen in Lyver et al. (2008) where they have successfully evaluated Māori harvesting practices using a qualitative approach and Gale et al. (2013) where the thematic analysis is described and justified as a useful and valid approach.
We believe that incorporating an additional quantitative approach is not necessary for this analysis because:
- Quantitative data cannot be collected from these Indigenous groups as practicing mutton-bird harvests remains illegal in their regions.
- Quantitative data cannot provide in-depth insight into to the motivations and values behind these cultural practices transmitted through oral traditions.
Reviewer 2 Report
This paper aims to obtain a better understanding of Māori natural resource management strategies from the cultural, spiritual, historical and ecological perspectives. However, this paper has flaws on the adopted methodology, and does not clarify the criticalness of the research value. The authors are suggested to revise this paper to enhance the criticalness of the study, questionnaire design, and the results interpretation. Detailed comments are shown as follows:
1. A significant portion of the Introduction section describes the background culture of the Māori people in New Zealand. However, little explanation is given regarding the research motivation, such as `why we currently need to investigate the Māori people, as opposed to other ethnic groups?' or `which aspects of previous studies on the Māori people have been less explored?'.
2. On the study purpose in the Introduction section, the discussion on the significance and representativeness of mutton-bird harvesting culture among the Māori people in New Zealand is lacking. What is the importance of exploring the Māori people's relationship between culture and the environment through the inquiry of this culture?
3. In the Materials and Methods section, the questionnaire design and background information of interview participants should be addressed to strengthen the relationship between the research purpose and the selection of interviewees.
4. In the Results and Discussion section, to provide readers with a more comprehensive understanding of the cultural significance of New Zealand's Māori people, relevant research findings should be reviewed. However, the authors' perspective on the relevant issues (including the cultural development of the Māori people, the impact of modern society on their culture, and the transmission of their cultural heritage) should be provided, and the relevant studies should be referred.
5. The last section should discuss the issues that require further investigation among the present-day Māori people, and elaborate on how this study relates to and impacts the development of the Māori community.
Author Response
Firstly, we would like to thank the reviewer for their time spent critiquing our manuscript. Specifically, we appreciate their insights regarding the context of the study and their suggested solutions to strengthen the manuscript’s critical evaluation of its research significance. Our responses to the comments are as follows:
- Research motivation (see lines 31-55). In summary, revitalisation of Indigenous cultural practices improves Indigenous and environmental wellbeing. Moreover, understandings of relational; environmental practice can improve resource management in Aotearoa New Zealand.
- Significance of harvests (see lines 166-174). Many of the world’s threatened seabird species are endemic to New Zealand. Utilising multiple approaches to manage their populations is important for global biodiversity.
- Further background as to why participants were selected has been added to the methods and materials section (see lines 129-137).
- Review relevant research in discussion (see lines 175-183). Previous work (both qualitative and quantitative have been incorporated.
- Future directions in conclusion (see lines 483-495). Māori practices and approaches to resource management should be incorporated into policy.
Reviewer 3 Report
The analysis here is of high quality, as is the writing. There is one minor issue and one major issue however. The minor issue is that it would be good to see more literature and theory associated with Indigenous environmental practices. Specifically, more general theoretical understandings on human abilities to create cultural "solves" for coordination problems to avert resource over-exploitation. (I am specifically thinking of Elenor Ostrom here). Perhaps:
Ostrom, E. (2000). Collective action and the evolution of social norms. Journal of economic perspectives, 14(3), 137-158.
It might be good to bolster the claim that Indigenous cultures are more nature-related by using evidence:
Niigaaniin, M., & MacNeill, T. (2022). Indigenous culture and nature relatedness: Results from a collaborative study. Environmental Development, 44, 100753.
That is the simple issue. The major problem is simply the number of participants interviewed. It appears that only two community members were interviewed here. I would want to see at least eight. Since the detail and nuance achieved with the original two interviews is high, additional interviews could be viewed as tools to verify the information already gained and to detect complexities or contradictions to these stories. That is to say that the paper could still rely on the two testimonies used throughout - but these should be verified by more interviews.
Given this I would not be comfortable recommending this for publication unless at least six more interviews were carried out. If these additional interviews were completed, however, I would have a great deal of confidence in this work. It may be possible for the author to use "snowball sampling" to find the additional individuals to interview through contacts already established. I would suggest this.
Author Response
The feedback given by this reviewer is greatly appreciated and we would like to thank them for their efforts. The suggestion develop our theoretical discussion further in terms of action was welcomed. We incorporated one of the suggested works (Niigaaniin & MacNeill, 2022) along with several others (e.g. Priadka et al., 2022; Skerrett and Ritchie (2021); Taiepa et al. (1997); and Toledo 2001) in both the introduction, discussion, and conclusion sections. After much consideration we chose to continue with these other chosen references instead of Ostrom (2000) as we felt they were more easily incorporated into the pre-existing discussion and examples used throughout the paper.
Regarding the major concern of the reviewer, our ability to respond to the request is limited. We acknowledge that the number of participants is very small for a publishable work. The main reasoning for this is the absence of harvesting practices in the regions where the study was conducted. For the last 2-3 generations, muttonbirding has been illegal and/or the local communities have been prevented from accessing muttonbirding sites (i.e. the selling of land to private land owners). As such, we only have the ability to turn to the very few who remember these old practices from their childhood. Addition of participants would risk the dilution of the data collected as their inputs would be based on assumption rather than direct success to oral tradition.
In this instance we feel this manuscript is an extraordinary case in that the quality of the data collected has not been impacted from a small sample size. Our confidence in the two participants chosen is based upon their status’ as identified knowledge holders by their people. As such they draw not only from personal experience, but the collective knowledge passed through generations of observation and practice. Therefore, their input is not theirs alone but reflective of a vast number of voices.
We hope that the editor and reviewer can empathise with this position and agree that the quality of data collected is sufficient to support this paper.
Reviewer 4 Report
Thank you for the opportunity to review this interesting and important paper that addresses Māori natural resource management focussing on the mutton-bird harvesting practices and associated cultural, spiritual, and environmental dimensions. In particular the paper introduces harvesting practices, explains the complexity of kaitiakitanga and associated governance and decision-making processes, and highlights the role of whanaungatanga emphasising collaboration in community relationships. The authors explain the links between Indigenous natural resource management and wellbeing of the community. This paper is placed within the emerging record of Indigenous natural resource management practices and focuses on the relationships between the Māori with mutton birds that warrant greater appreciation and incorporation in formal management practices.The paper as it is well-written, clear, well-structured, and focused and I don’t have any significant issues to highlight.
My only minor comment for the authors would be to provide suggestions for future research on the topic, and some more guidance on how to pursue the "change in worldview" and any policy-focused recommendations for the "revitalization of customary practices" in New Zealand.
Some minor grammar edits are required on line 64 "is intertwined" (add with?) and 406 "that respects and reciprocity". I suggest authors conduct final proofreading of the article.
Author Response
We would like to thank this reviewer for their feedback and support in this paper. We have amended the highlighted grammatical errors and completed a final proofread of the article as requested.
Round 2
Reviewer 2 Report
This paper has been revised according to my previous comments. Therefore, it is suggested that it can be accepted after typos and grammers check.
Author Response
A final proofreading of the manuscript has been completed
Reviewer 3 Report
I appreciate the changes made to the literature review and theoretical frame. I also appreciate the explanation that, essentially, the two participants are some of the very few, perhaps the only, remaining group members with direct experiential knowledge of these traditional practices. In addition to the explanation given in the authors' response, I think there is a point to be made that the knowledge of these two specific people offers perhaps a last chance to try and understand traditional knowledge before it is lost. That contains immeasurable value despite the low participant number.
I think the authors have made that case fairly well in the response, but not in the paper itself. This is a lost opportunity to "sell" the importance of the research to potential readers. I would suggest that the authors add a short paragraph in the methods section describing the rarity of the knowledge (and associated people) that are participating. And reiterate the importance of documenting such knowledge before it disappears, or even as a way of preventing it from disappearing. The author/s may also want to make a note in both the introduction and abstract to this effect. It can change a perceived weakness to an important strength.
Author Response
Thank you once again for your input to improve our manuscript. We agree that not including this argument in the manuscript was an oversight on our part. We have incorporated these points into the methods section (see lines 142-149).